# Research on Behavior Recognition and Online Monitoring System for Liaoning Cashmere Goats Based on Deep Learning

**DOI:** 10.3390/ani14223197

**Published:** 2024-11-07

**Authors:** Geng Chen, Zhiyu Yuan, Xinhui Luo, Jinxin Liang, Chunxin Wang

**Affiliations:** 1Animal Husbandry and Veterinary Research Institute, Jilin Academy of Agricultural Sciences, Shengtai Street, Changchun 130033, China; cg907287301@163.com (G.C.); yuanzhiyu2019@163.com (Z.Y.); luoxinhui615@sina.com (X.L.); liangjinxin0124@163.com (J.L.); 2College of Animal Science and Technology, Jilin Agricultural University, Xincheng Street, Changchun 130118, China

**Keywords:** deep learning, Liaoning Cashmere Goat, YOLOv8n, behavior recognition

## Abstract

The Liaoning Cashmere Goat is a valuable breed in China, known for its cashmere and meat. To enhance intensive farming efficiency, we developed an intelligent behavior recognition system using deep learning. We compared Faster R-CNN and YOLO algorithms and found YOLOv8n achieved an average accuracy of 95.31% within 50 epochs. Our improvements led to a detection accuracy of 98.11%. The online detection system offers real-time monitoring, reducing manual labor and improving animal welfare, benefiting the livestock industry.

## 1. Introduction

Liaoning Cashmere Goat is a variety of cashmere goat formed by long-term artificial domestication and careful cultivation, which has the characteristics of high cashmere yield, with fine and long cashmere fiber [1]. In the process of intensive breeding, the behavior of goatsis a comprehensive reflection of the adaptation degree and health status of the breeding environment. The study of the behavior of goats is of great significance to improve the enclosure environment, prevent and control diseases, and improve the production performance of goats [2]. At present, goat behavior recognition mainly relies on manual work, which is inefficient, time-consuming, labor-intensive, and difficult to monitor around the clock. Therefore, this study designed and built an online detection system for efficient identification of goat behavior.

In recent years, with the advancement of smart agriculture, the identification of goat behavior has garnered widespread attention from researchers. Currently, goat behavior recognition is primarily conducted using two methods: one is contact-based identification methods using sensors, including three-dimensional accelerometers and acoustic sensors; the other is non-contact computer vision-based recognition methods. Both approaches have yielded a series of research outcomes, providing robust technical support for the development of precision livestock farming.

In contact-based identification methods, the recognition of goat behavior based on three-dimensional accelerometers has been applied to the study of identifying behaviors such as active, inactive, grazing, standing [3], walking [4], and parturition [5]. These accelerometers are typically mounted on the goat’s legs or back, or are integrated into electronic ear tags and collars to collect acceleration change information during different behavioral states. By matching these acceleration data with concurrent observational records, researchers can utilize machine learning and deep learning techniques, such as Quadratic Discriminant Analysis (QDA), Linear Discriminant Analysis (LDA), convolutional neural networks (CNNs), and Transfer Learning (TL), for accurate identification and classification of goat behavior. Acoustic sensors are mainly applied to the recognition of foraging and estrus behaviors in sheep [6]. Researchers install devices integrated with acoustic sensors on goats to capture sound signals, and through data processing techniques to enhance signal quality, they then employ Support Vector Machines (SVMs) and deep learning for training behavioral recognition models.

Non-contact identification methods leverage image processing and machine learning technologies to analyze video or image data, thereby enabling precise counting of goat populations and accurate recognition of individual behaviors [7]. These methods overcome the issues of stress and nibbling associated with direct contact sensors on goats, allowing for continuous behavioral monitoring without disrupting the normal activities of the animals [8].

Computer vision has been preliminarily applied in the field of goat inventory and behavior recognition. Gu et al. [9] employed a two-stage object detection algorithm to detect six types of behaviors in goats: standing, feeding, lying, attacking, biting, and climbing. The mean average precision (mAP) for these two categories of behaviors exceeds 98%. In the classification stage, the accuracy for all behaviors surpasses 94%. The model’s memory footprint is less than 130 MB. Yu et al. proposed two algorithmic models for the detection of estrus behavior in ewes on large-scale meat goat farms. The first, the addition of two residue units to the final residue structure of YOLOv3 and the clustering K-means + + algorithm, yielded a model size of 276 MB, an accuracy of 98.56%, and a recall of 98.04% [10]. The second is to replace the backbone feature extraction network Darknet-53 of YOLOv3 with a more lightweight EfficientNet-B0 and add attention module SENet in MBConv. The model size is 40.6 MB, with 99.44% accuracy and a recall of 95.5% [11]. Based on the YOLOv4 model, target tracking algorithm, and decision tree algorithm, Gonzalez-Baldizon et al. [12] proposed a method to automatically identify the abnormal behavior of lambs when they are fed under closed conditions. The above method only realizes the behavior detection of a single goat, and the behavior identification of group-fed goats requires the integration of various algorithms and large computing power, which cannot meet the needs of intensive breeding and production sites. Therefore, deep learning technology can provide a non-contact detection method for flocks of goats. However, current detection models are relatively complex with a large number of parameters and there is a lack of practical application in goat farming environments.

This study focuses on the research of a deep learning-based algorithm for the behavioral recognition of Liaoning Cashmere Goats. The aim is to utilize deep learning algorithms to achieve automatic recognition and analysis of the behaviors of Liaoning Cashmere Goats, thereby assisting in addressing practical issues in the breeding process, such as the lack of timely medical treatment due to inadequate supervision, dystocia, improper care for newborn kids, and loss of goat assets.

The main contributions of this study are summarized as follows:We trained common object detection algorithms using the Liaoning Cashmere Goat behavior recognition dataset and determined that YOLOv8n is suitable as a baseline algorithm for the behavior recognition of Liaoning Cashmere Goats.We designed a vertical image augmentation method through splicing and applied the CLAHE technique to enhance nighttime images, increasing image detail and reducing the difficulty of dataset annotation.We improved the algorithm based on SlimNeck and GhostNet, and employed various attention mechanisms and loss functions to refine the YOLOv8n algorithm, resulting in three more lightweight and accurate models.Building upon the Liaoning Cashmere Goat behavior recognition model, we constructed an online monitoring and behavior recognition system for Liaoning Cashmere Goats.We designed a human–machine interaction interface and behavior recording form for the system and verified the accuracy of the system’s records in real scenarios.

The remainder of this paper is organized as follows. We provide a comprehensive review of the object detection algorithm and its improvement methods in Section 2. We introduce the original dataset and select the baseline algorithms through preliminary experiments in Section 3. We describe the experimental plan for the algorithm model in Section 4. We present the results and analysis of the algorithm improvement experiments in Section 5. We introduce the composition of the Liaoning Cashmere Goat behavior recognition and online detection system and conduct tests on the system in Section 6. We summarize the current work and provide an outlook on future work in Section 7.

## 2. Related Work

### 2.1. Object Detection Algorithm

Object detection is one of the most important tasks in the field of computer vision, with the primary goal of automatically detecting and localizing target objects in input images or videos. The development history of the object detection algorithm is shown in Figure 1.

In object detection, two-stage and one-stage are the two primary approaches. Two-stage object detection algorithms include R-CNN, SPP-Net, Fast R-CNN, Faster R-CNN, and R-FCN, among others [13]. These algorithms typically generate a set of candidate boxes (also known as region proposals) first and then employ convolutional neural networks (CNNs) to classify and localize these proposals. While these algorithms generally offer higher accuracy, they correspondingly require more computational resources and time. Common one-stage object detection algorithms include OverFeat [14], YOLOv3, YOLOv4 [15], YOLOv5, YOLOX, SSD [16], and RetinaNet. These algorithms are generally faster and require fewer computational resources, but they tend to have lower accuracy. In recent years, with the continuous development of technology, some new one-stage object detection algorithms, such as YOLOv5 and YOLOX, have achieved accuracy comparable to that of two-stage algorithms. This study aims to compare the typical two-stage and one-stage object detection algorithms to establish a baseline algorithm suitable for the behavior recognition of Liaoning Cashmere Goats.

### 2.2. Nighttime Image Enhancement in Goat Pen

Image enhancement is a critical step in image processing, which is especially important when dealing with low-light images in goathouses at night, where even the human eye cannot accurately observe the behavior of each goat. Traditional image contrast enhancement methods, such as linear stretching and gray-level transformation, may lead to the loss of important details. Histogram equalization (HE) is a nonlinear method that enhances global contrast by adjusting the image’s histogram distribution but may result in the loss of detail information in high-contrast areas [17]. To address this issue, Contrast-Limited Adaptive Histogram Equalization (CLAHE) was proposed. It segments the image into multiple small regions (tiles), performs histogram equalization independently in each tile, and merges the results using bilinear interpolation to improve contrast while preserving local details [18]. The performance of CLAHE is influenced by two key parameters: the number of tiles and the clip limit, where tile size determines the segmentation of the image, and the clip limit controls the contrast to prevent saturation in uniform areas [19]. Due to its advantages in improving image contrast and preserving details, CLAHE has been widely applied in fields such as medical imaging, satellite image processing, and video enhancement, particularly in medical imaging, where it assists doctors in more accurately identifying and diagnosing diseases [20]. This study intends to use CLAHE for the enhancement of nighttime images in goat houses to assist researchers in annotating datasets and improving the accuracy of goat behavior recognition.

### 2.3. Attention Mechanism

The Squeeze-and-Excitation (SE) attention mechanism introduces channel attention to the model learning process, adjusting the weights of different channels through learning, thereby enhancing the expressive power of convolutional neural networks [21]. The Convolutional Block Attention Module (CBAM) is an attention module that combines channel attention and spatial attention, which is capable of modeling inter-channel relationships and spatial locations simultaneously, thus effectively enhancing the model’s ability to represent image features [22]. Applying the CBAM attention mechanism to the Feature Pyramid Network (FPN) of YOLOv8 can further enhance the model’s expressive and discriminative capabilities, aiding in the improved accuracy of behavior recognition for Liaoning Cashmere Goats. The non-local attention mechanism captures long-range and global dependencies, strengthening the model’s perception of global information in images [23]. By reinforcing the interaction and dependency between feature points, this attention mechanism can better understand and distinguish the subtle differences between different behaviors. The non-local attention mechanism, through its enhancement of interaction and dependency between feature points, can assist the model in better understanding and differentiating the subtle differences between various behaviors [24]. Therefore, this study introduces the SE, CBAM, and non-local attention mechanisms into the FPN of YOLOv8, comparing their effects to explore the most suitable attention mechanism for improving the recognition effectiveness of Liaoning Cashmere Goat behaviors.

### 2.4. Loss Function

The loss value of YOLOv8 is composed of two parts: regression loss and class loss. Additionally, since YOLOv8 employs the Deconvolutional Feature Learning (DFL) model as the regression predictor, the regression component includes not only the calculation of the Intersection over Union (IOU) loss but also the DFL loss. The class loss is derived from the cross-entropy loss calculated based on the predicted class of the prior box and the true class of the bounding box. To further enhance the object detection performance of YOLOv8, this study explores four methods: Complete Intersection over Union (CIOU) [25], Enhanced Intersection over Union (EIOU) [26], Scaled Intersection over Union (SIOU) [27], and Alpha-weighted Complete Intersection over Union (Alpha-CIOU) [28] that offer more accurate regression loss calculations compared to the traditional IOU method, and compares their effects to improve the model’s accuracy in detecting the behaviors of Liaoning Cashmere Goats.

## 3. Preliminaries

### 3.1. Faster-RCNN and the YOLO Series Algorithm

Faster R-CNN is a widely used object detection algorithm that can be divided into two main components: initially, a feature extraction network is employed to extract features from the input image, followed by the utilization of the Region Proposal Network (RPN) to swiftly generate candidate object bounding boxes, with final curation and localization of these boxes performed through classification and localization tasks. Compared to previous object detection algorithms, the uniqueness of Faster R-CNN lies in its ability to share the computational outcomes of the feature extraction network, thereby avoiding redundant calculations and reducing computational costs. Additionally, the use of the RPN network to rapidly produce candidate bounding boxes enhances both detection speed and accuracy. Faster R-CNN comprises four network modules: the feature extraction network, the RPN network, the Region of Interest (ROI) Pooling layer, and the classification and localization network.

You Only Look Once (YOLO) is a full-end object detection approach that simplifies the detection task into a regression problem. It predicts the category and location of objects through a deep neural network in a single forward pass, generating bounding boxes with confidence scores for each grid in the image. From YOLOv1 to YOLOv5, the YOLO algorithm has undergone multiple generations of evolution, continuously innovating and optimizing in various aspects such as network architecture, training processes, data augmentation, and attention mechanisms, with the goal of enhancing detection accuracy while ensuring real-time processing capabilities. A notable feature of YOLO is its combination of speed and precision, enabling real-time detection and localization of multiple objects in an image. In February 2023, Ultralytics introduced the YOLOv8 algorithm, based on YOLOv5. As the latest variant of the YOLO series, the model is divided into three main parts: the backbone, the neck, and the detection head. The overall structure of the YOLOv8 model is depicted in Figure 2. This modular design allows the YOLOv8 algorithm to flexibly adapt to different application scenarios and has demonstrated excellent performance in experiments.

### 3.2. Dataset Construction and Training Configuration

#### 3.2.1. Keyframe Extraction

The behavioral videos of Liaoning Cashmere Goats were collected at a breeding center for Liaoning Cashmere Goats. The experimental goat pens were semi-open with windows and an outdoor exercise area, allowing the goats free access in and out. The goats were fed using a combination of manual feeding and free access to water under a fully housed rearing system. The rearing density was set at $2\;m^{2}$ per goat for the high-density group and $3\;m^{2}$ per goat for the low-density group. Sixteen high-definition cameras were used to record the activities within the pens. The cameras were installed at the highest point on the inner corner of the goat pens, 2.5m above the ground, providing an overview of the entire area. The cameras had a focal length of 6 mm, a frame rate of 30FPS, and a resolution of 1920×1080pixels. In this study, Dahua HDCVI series digital video recorders and SmartPlayer video players were used to convert DAV format video files into AVI format, which were then converted into MP4 files suitable for image recognition using video editing software. Subsequently, Python was employed to write a keyframe extraction script, from which a total of 1500 images were extracted from the videos recorded over three months in 16 goat pens for the creation of the dataset [29]. The original images collected under various rearing densities and lighting conditions [30] are shown in Figure 3.

#### 3.2.2. Informed Consent Statement

Informed consent was obtained from all subjects involved in the study. For studies involving client-owned animals, written informed consent was obtained from the owners or their authorized agents.

#### 3.2.3. Behavioral Annotation

In this study, we initially utilized the LabelImg tool to annotate the distinct behaviors exhibited by goats. within the images, such as lying, standing, drinking, and eating, to create a Pascal VOC dataset for goat behavior recognition. A total of 8418 goats were annotated, with the following breakdown of labels: 4415 for lying, 2869 for standing, 112 for drinking, and 1022 for eating. The dataset, after annotation of the original data, is illustrated in Figure 4.

#### 3.2.4. Experimental Parameter Settings

In this study, the learning rate was set to 0.001, with the Adam optimizer employed for convergence. Regularization was performed using L2 regularization, with a weight decay coefficient of 1 × $10^{-5}$. Additionally, the model utilized the Mixup data augmentation technique during training to enhance its generalization capabilities [31].

#### 3.2.5. Training Machine Configuration

The training machine used in this experiment is equipped with an NVIDIA GTX 2080Ti graphics processing unit (GPU), featuring 11 GB of dedicated video memory. The host is configured with 14 GB of RAM and an 8-core Intel® Xeon® Gold 6130 CPU. Model development and training were conducted on the Windows 10 operating system using Python 3.8 and PyTorch 1.8. The configuration of the training machine is detailed in Table 1.

#### 3.2.6. Algorithm Evaluation Metrics

Mean Average Precision. To assess the accuracy of the two object detection algorithms, this study adopts the mean average precision (mAP) as the primary evaluation metric. This metric takes into account the precision and recall of the model at various confidence thresholds and evaluates the average performance across all categories.Detection Frame Rate. The detection frame rate, measured in frames per second (FPS), reflects the time required for the model to process each frame of an image. In practical applications, a higher detection speed is essential to meet the demands of real-time processing.Number of Parameters. The number of parameters in a model reflects its storage requirements and computational complexity. This study records and analyzes the total number of parameters in the model to assess its scale and deployability.

#### 3.2.7. Behavioral Recognition Results and Analysis

In this study, the Liaoning Cashmere Goat behavior recognition dataset was utilized to train the Faster R-CNN and the YOLO series of algorithms. After every 10 epochs of training, a rapid assessment of the mAP value was conducted on the validation set using a subset of predicted bounding boxes, with the results depicted in Figure 5.

As can be seen from Figure 5 in the task of behavior recognition for Liaoning Cashmere Goats, YOLOv8n achieved convergence around the 50th epoch, demonstrating the fastest convergence rate and the highest detection accuracy.

Upon completion of 200 training epochs, the mAP values for each behavior were calculated for YOLOv8n with a confidence threshold of 0.001 and a non-maximum suppression threshold of 0.5, as shown in Figure 6.

Figure 6 indicates that the YOLOv8n algorithm achieved an average detection accuracy of 95.31% on the original dataset of Liaoning Cashmere Goat behavior recognition. Consequently, this study adopts YOLOv8n as the baseline model for further refinement and utilizes it for the construction of the Liaoning Cashmere Goat behavior recognition system.

In this section, we compared the convergence speed and detection accuracy of the typical two-stage object detection algorithm Faster R-CNN and the typical one-stage object detection algorithm YOLO in the task of behavior recognition for Liaoning Cashmere Goats. We found that the YOLOv8n algorithm outperforms Faster R-CNN and other algorithms in the YOLO series for this task. The YOLO algorithm is well suited as a baseline for further refinement in the improvement of behavior recognition algorithms for Liaoning Cashmere Goats.

## 4. Experiments

### 4.1. Use of Data Augmentation Methods

#### 4.1.1. CLAHE Enhancement Method

Goat housing nighttime images often suffer from low light conditions, resulting in poor contrast that poses challenges for image annotation and hinders the learning of image features by algorithms. Therefore, this study employed Contrast Limited Adaptive Histogram Equalization (CLAHE) to enhance nighttime images. The contrast limit was set to 4, and the grid size was configured at 16×16. The enhancement effect is shown in Figure 7.

From Figure 7, it can be seen that the adaptive histogram equalization method effectively improves image contrast, facilitating the creation of the dataset and aiding in the learning of more image features during algorithm training. This study quantitatively evaluated the nighttime image set before and after CLAHE enhancement using two quantitative metrics: edge-based contrast measurement (EBCM) and discrete entropy (DE) [19]. The results are depicted in Figure 8.

Figure 8 indicates that for the Liaoning Cashmere Goat behavior recognition nighttime image set, there is a highly significant (p<0.01) increase in the edge contrast measurement and discrete entropy values before and after CLAHE enhancement. This suggests that CLAHE can enhance the details, textures, and information content of the nighttime images for Liaoning Cashmere Goat behavior recognition.

#### 4.1.2. Image Stitching

The Mosaic data augmentation algorithm is a technique that involves randomly cropping, scaling, and then stitching four images together to form a single composite image [32]. The learning gain from a single synthesized image is quadruple that of the original images [33]. In practical tasks, the target object is not always fully presented within the image; it may sometimes be occluded or incomplete. The application of the Mosaic data augmentation method to the Liaoning Cashmere Goat behavior recognition dataset is illustrated in Figure 9.

Due to the potential for the Mosaic data augmentation method to cut through the subjects, such as goats, resulting in incomplete labels, this study designed a vertical stitching method for data augmentation. This simple yet effective method not only maintains scene continuity and preserves the integrity of the images but also avoids misclassification of negative samples. By simulating different monitoring perspectives and distances, the vertical stitching method enhances the model’s ability to recognize goat behavior under varying breeding densities and lighting conditions. This augmented dataset approach is shown in Figure 10.

### 4.2. Algorithm Lightweighting

In order to maintain model accuracy while enhancing its lightweight properties, this study integrated the residual stacking structures of GhostNet and SlimNeck into the C2f module of the backbone and neck networks, devising the GC-C2f and SC-C2f residual structures. This is depicted in Figure 11.

#### 4.2.1. GC-C2f

This study designed the GC-C2f based on the GhostNet architecture (see Figure 11c) by integrating channel attention (CA) into the Ghost Module to construct the GC-BottleNeck (see Figure 11b), which replaced the BottleNeck in the C2f of the YOLOv8 model’s backbone and neck (see Figure 11a) [33]. This approach facilitates the generation of a multitude of feature map structures during feature extraction, thereby producing a complete set of output channels with reduced computational requirements, achieving the goal of model lightweighting [34].

#### 4.2.2. SC-C2f

The SC-C2f was designed in this study based on SlimNeck architecture (see Figure 11f), by embedding channel attention (CA) into the GSConv structure composed of depthwise separable convolutions (DSCs), creating the SC-BottleNeck convolutional structure (see Figure 11e) for processing multi-channel feature mapping information. This convolutional structure then replaced the BottleNeck in the C2f module [35]. Utilizing this method allows the GSConv convolution (see Figure 11d) to maintain the correlation between channels, mitigating the impact on model accuracy associated with the use of depthwise separable convolutions [25].

#### 4.2.3. Hybrid Lightweighting with GC-C2f and SC-C2f Residual Structures

Considering that GhostNet can effectively reduce the computational complexity of the model, it can potentially provide computational headroom for SlimNeck to handle additional features, thus aiding the model in achieving superior performance. In this study, the C2f in the backbone was replaced with GC-C2f, and the C2f in the FPN was replaced with SC-C2f, forming a new lightweight model.

## 5. Comparison and Results Analysis

### 5.1. Comparison of Data Augmentation Effects Between Mosaic Method and Vertical Splicing Method

This study employed the YOLOv8n algorithm to train datasets enhanced by both the Mosaic method and the vertical splicing method, with the outcomes presented in Figure 12.

As depicted in Figure 12, the average recognition accuracy achieved using the Mosaic data augmentation method was 95.12%, which is lower than the 95.31% accuracy obtained with the original dataset as described in Section 3.2.7 of this paper. In contrast, the vertical splicing method utilized in this study increased the average recognition accuracy by 1.68% relative to the original dataset, reaching 96.99%. This enhancement may be attributed to the severe occlusions among the Liaoning Cashmere Goats in the behavior recognition dataset; the Mosaic method can lead to the cutting of goats, resulting in the loss of labels and the creation of negative samples, which is detrimental to the algorithm’s learning of behavioral features. The vertical splicing method, however, avoids these shortcomings and increases the number of targets within each image, enabling the algorithm to learn a greater array of behavioral characteristics. Consequently, the vertical splicing method is deemed more suitable for data augmentation in Liaoning Cashmere Goat behavior recognition than the Mosaic data augmentation method.

### 5.2. Attention Mechanism and Loss Function Improvement Results

Building upon the lightweight refinement of the YOLOv8n algorithm based on SlimNeck, this study initially integrated four attention mechanisms into the algorithm’s Feature Pyramid Network (FPN): Channel Attention (CA), Squeeze-and-Excitation (SE), Convolutional Block Attention Module (CBAM), and non-local attention. Subsequently, in the detection head section, three loss functions, Enhanced Intersection over Union (EIOU), Scaled Intersection over Union (SIOU), and Alpha-weighted Complete Intersection over Union (Alpha-CIOU), were introduced and compared with the original Complete Intersection over Union (CIOU). The model’s mean average precision (mAP), memory occupancy, and parameter count are presented in Table 2.

### 5.3. Ablation Study

To assess the effectiveness of each improvement step, this study conducted an ablation study, systematically dividing the adjustments to YOLOv8n into three distinct phases. In the first phase (Step 1), the model was refined for lightweight modifications using SC-C2f, GC-C2f, and a hybrid approach combining both SC-C2f and GC-C2f. The second phase (Step 2) applied the CBAM attention mechanism within the FPN. The third phase (Step 3) utilized the Alpha-CIOU in the detection head. The results are presented in Table 3.

From Table 3, it can be observed that in the first step of improvement, the incorporation of GC-C2f, SC-C2f, and a hybrid of GC-C2f and SC-C2f resulted in decreases in the mean average precision (mAP) of the model by 1.27%, 1.54%, and 1.10%, respectively, and reductions in memory usage by 29.0%, 18.9%, and 28.9%, respectively. When employing GC-C2f for lightweight improvements to the model, the reduction in parameters can be attributed to GSConv’s ability to mimic the output of Standard Convolution (SC) while reducing computational load, effectively decreasing the number of parameters and memory usage without significantly compromising detection accuracy. Utilizing the Ghost module in GC-C2f for lightweight enhancements, a convolutional layer is decomposed into two smaller layers: a primary convolutional layer and a ghost convolutional layer. The ghost convolutional layer computes the output using a subset of the channels from the original layer, while the primary convolutional layer utilizes the remaining channels. This decomposition significantly reduces the computational burden and the number of parameters, thereby decreasing memory usage. However, the SlimNeck and GhostNet structures are designed to reduce model complexity, leading to a reduction in parameters and memory usage. This reduction in complexity may result in a slight decrease in the model’s ability to capture fine details, thereby affecting the mean average precision (mAP).

In the second step of enhancement, applying the Convolutional Block Attention Module (CBAM) to the Feature Pyramid Network (FPN) led to increases in the mAP by 1.68%, 0.68%, and 1.80%, respectively. The third step involved optimizing the loss function calculation method by employing the Alpha-weighted Complete Intersection over Union (Alpha-CIOU), which further improved the model’s mAP by 0.45%, 1.07%, and 0.50%. Additionally, introducing attention mechanisms increases the model’s memory consumption and parameter count, whereas modifying the loss function does not have these effects.

### 5.4. Behavioral Recognition Results

To verify the superiority of the three models proposed in this study for recognizing the behaviors of Liaoning Cashmere Goats, ”stand”, “eat”, “lying”, and “drink”, we compared our methods with the original YOLOv8n algorithm. The comparative results are detailed in Table 4.

From Table 4 it is evident that the mAP for the four behaviors of Liaoning Cashmere Goats detected by the three improved models all exceeded 97%, and they improved upon the YOLOv8n by 0.86%, 0.22%, and 1.12%, respectively. This indicates that the adverse impact on model accuracy due to the lightweighting methods in this study has been mitigated by the incorporation of attention mechanisms and the optimization of the loss function. Among the recognition of the four behaviors by YOLOv8n, the standing behavior had the lowest recognition accuracy of 94.45%. The analysis attributes this to the herding behavior of goats, where frontal and rear occlusions are more severe for standing behaviors, thereby making it difficult for the model to obtain effective features. However, the mAP for the standing behavior recognition of Liaoning Cashmere Goats by the three improved models increased by 1.52%, 0.20%, and 1.83% over the original model. This suggests that the improved methods have enhanced the recognition capability for occluded targets.

### 5.5. Recognition Results in Shedding of Varying Densities

To ascertain the efficacy of the models developed in this chapter for the behavioral recognition of Liaoning Cashmere Goats under varying rearing densities, the model with the highest precision, GSCA-YOLOv8n, was utilized for testing. The tests were conducted on groups with rearing densities of $2\;m^{2}$ per goat for the high-density group and $3\;m^{2}$ per goat per goat (low-density group). The outcomes of the tests are illustrated in Figure 13, demonstrating recognition rates of 98.14% and 98.27%, respectively.

From Figure 13, it is evident that the model’s recognition accuracy is lower for high-density housing compared to low-density housing, indicating that the model’s capability to handle tasks with densely packed targets is relatively poor. The vertical splicing method designed in this study can be considered as placing more goat targets in the 640×640pixels input images, which is beneficial for the model to learn more features of the goats. Additionally, before the vertical splicing, the size of the original images processed by the editing software was 1280×720pixels, and after splicing, it became 1280×1440pixels, which is closer to the square image input format required by the YOLOv8n data preprocessing. This means that the proportion of gray background that needs to be filled during image preprocessing is less than that of the original image [36]. This also allows the convolutional kernels used to learn edge information of the image to obtain more real image information, rather than gray-filled information, which contributes to improving the quality of feature learning and the ultimate performance of the model.

### 5.6. Comparison with State-of-the-Art (SOTA) Models

To validate the effectiveness of the three models proposed in this study for goat behavior recognition, the improved recognition methods were compared with Centernet, Faster R-CNN, SSD, YOLO v7-tiny, and YOLO v9-s [37], among others. The YOLO series represents the current mainstream object detection algorithms, while Centernet [38], Faster R-CNN, and SSD [39] are algorithms that have demonstrated good performance in other tasks. To verify the superiority of the methods in this study, the same configuration of training machines was used to train each algorithm. The training results are depicted in Figure 14.

From Figure 14, it can be observed that the three models developed in this study outperform other state-of-the-art (SOTA) object detection algorithms in terms of recall rate, precision, and mean average precision (mAP). The improved methods demonstrate mAP increases of 27.3%, 26.7%, and 27.5% over SSD, respectively, confirming the superiority of our approaches in terms of recognition accuracy for Liaoning Cashmere Goat behavior.

To further validate the suitability of our methods for deployment on edge devices, this study compared the memory usage and frames per second (FPS) of the three models with those of YOLO v5s, YOLO v7-tiny, YOLO v9-s, SSD, Centernet, and Faster R-CNN. The comparative results are presented in Table 5.

From Table 5, it is evident that the three methods developed in this study consume less memory and have a lower parameter count compared to other object detection algorithms. However, their detection speeds do not match those of SSD and Faster R-CNN. Notably, the detection speed of GSCA-YOLOv8n is only 13.24 FPS, which may be attributed to the use of depthwise separable convolutions in various positions within the network architecture, potentially hindering the high-performance GPU from fully leveraging its parallel computing capabilities [40].

### 5.7. Comparison of Attention Heatmaps

To investigate the role of different attention mechanisms in goat behavior recognition, this study recorded heatmaps reflecting the attention applied within the model’s Feature Pyramid Network (FPN) using various attention mechanisms, as depicted in Figure 15.

As can be observed from Figure 15, the heatmaps generated by the four attention mechanisms do not exhibit significant differences in the left and central areas of the goat pen. However, non-local attention and Convolutional Block Attention Module (CBAM) enable the model to focus more on regions with a higher density of standing goats (such as the upper-right corner of the image). Therefore, when selecting an attention mechanism, it is necessary to choose and integrate different attention mechanisms sensibly in accordance with the characteristics of the dataset and in conjunction with the network architecture.

### 5.8. Algorithm Improvement Summary

This chapter presents improvements to the YOLOv8n algorithm across four key aspects: data augmentation, algorithm lightweighting, incorporation of attention mechanisms, and refinement of the loss function. The enhanced models were compared with the baseline model, as well as with other state-of-the-art (SOTA) object detection algorithms, including Centerent, SSD, Faster R-CNN, and YOLOv7-tiny. Three improved methods suitable for the behavior recognition task of Liaoning Cashmere Goats were obtained.

Among them, the lightweighting method based on SlimNeck, which resulted in SNCA-YOLOv8n, demonstrated the best performance in detection speed; it was capable of achieving 31.4 FPS. The lightweighting method based on GhostNet, leading to GNCA-YOLOv8n, had a weight file size of 8.9 MB and a parameter count of 2.21 M, offering a more significant advantage in terms of lightweighting. Furthermore, by employing GhostNet in the backbone feature extraction network of YOLOv8n and SlimNeck in the neck to strengthen feature extraction, GSCA-YOLOv8n was developed, with an mAP of 98.11%, which is the highest among the SOTA object detection algorithms.

## 6. Development of an Online Detection System for Liaoning Cashmere Goat Behavior Recognition

### 6.1. System Structure and Design

The composition of the online detection system for Liaoning Cashmere Goat behavior recognition is depicted in Figure 16. The system consists of a video acquisition module, a video processing module, a behavior detection and recognition module, and a human–machine interaction module. The video acquisition module, composed of a network camera and mobile WiFi, facilitates the real-time collection and upload of video information from the goat pen. The video processing module is responsible for sampling and enhancement of the video. The detection and recognition module achieves the identification of goat behavior in the images, including parameters such as behavior type, quantity, and frame rate. Finally, the human–machine interaction module is in charge of visualizing and presenting the data.

### 6.2. Video Acquisition Module

Due to the large space and complex structure within the goat pen, this study introduced wireless panoramic network cameras and outdoor full-color wireless ball machines to obtain real-time video information from both inside and outside the goat pen. The device models are shown in Table 6.

At a breeding center for Liaoning Cashmere Goats, the wireless panoramic camera is installed on a beam at a height of 3.7 m in the center of the goat pen, ensuring a comprehensive view of the entire pen from an overhead perspective. The field of view of the wireless panoramic network camera is shown in Figure 17a,b. The outdoor full-color wireless ball machine is installed at a height of 2.5 m on a corner of the fence, ensuring a comprehensive view of the entire field. This device is equipped with waterproof and dustproof capabilities, ensuring continuous operation even under adverse weather conditions, such as low temperatures, rain, snow, and sandstorms. The installation location and camera footage of the camera are depicted in Figure 17c,d.

Field tests have shown that the built-in 128 GB SD card of the camera can store 15 days of activity video (the camera automatically deletes static frames) or 6 days of continuous video (the camera does not automatically delete any video segments).

### 6.3. Video Processing Module

The video processing module is responsible for the transmission and sampling of collected video data, performing preliminary treatments to provide optimized and stable input data for subsequent behavior detection and recognition modules. In this study, wireless WiFi is utilized to provide network transmission for the wireless panoramic network cameras inside the goat pen and the full-color wireless ball machines outside. The screen information is captured using the Python mss package to obtain video footage. Employing this method for real-time reading of Liaoning Cashmere Goat farm videos can assist the video processing module in acquiring higher quality real-time video. As shown in Figure 18, the use of adaptive histograms for data enhancement did not significantly reduce the frame rate of the video stream, with 83.6% of the time maintaining a frame rate that exceeds the detection speed of SNCA-YOLOv8n, ensuring no adverse impact on the performance of the algorithm.

### 6.4. Behavior Detection and Recognition Module

The behavior detection and recognition module is the core component of the Liaoning Cashmere Goat online system. In this study, the SNCA-YOLOv8n, GNCA-YOLOv8n, and GSCA-YOLOv8n behavior recognition models obtained in Section 4 are utilized for the online behavior detection of Liaoning Cashmere Goats. The system operation modes are set at high-speed detection mode with SNCA-YOLOv8n as the core detection algorithm, balanced detection mode with GNCA-YOLOv8n as the core detection algorithm, and precision detection mode with GSCA-YOLOv8n as the core detection algorithm.

The parameters for behavior recognition include the number of goats standing, drinking, lying down, and feeding in a single frame, which are printed at the terminal. Labeling and the corresponding number of goats are recorded from the start of the program operation, and the user is prompted to save the quantity information of each behavior upon completion of the run. The quantities of each behavior are recorded at a frequency of one set of data per second, as shown in Table 7.

### 6.5. Human–Machine Interaction Module

The human–machine interaction module is primarily responsible for interfacing with hardware and visualizing data, allowing users to control the loading of images and videos through the interface. While dynamically displaying images and video information, the detected and recognized parameters are concurrently presented in a list format. This system is written in PyQt5, and the human–machine interaction interface is shown in Figure 19.

The aforementioned modules constitute a complete online detection system framework, providing an analytical solution for the behavior recognition of Liaoning Cashmere Goats. The online operation effect of the system is illustrated in the following Figure 20. On the left, log in to the monitoring platform and open the live scene page. After the system captures the screen information from the left computer, it displays the recognition results on the right screen after detection and identification.

### 6.6. Human–Machine Interaction Module

To verify the stability of this system, this study conducted recognition and recording of a 2 h online goat farm video. The recording method involved separately recording the total time for standing, feeding, drinking, and lying down behaviors of 28 goats in the pen. The comparison between system recognition and manual recognition is shown in Figure 21.

Data from Figure 21 indicate that when recording the feeding behavior of Liaoning Cashmere Goats, the system-identified feeding time was 6.3% longer than that identified manually. However, the sum of feeding and standing behaviors differed by only 1% between the two detection results. This discrepancy arises because goats may have their heads in the feed trough without actually eating, which is marked as feeding during behavior annotation. This does not align with the manual recognition results, thus causing recognition errors. Moreover, when goats huddle together to feed, the mutual occlusion between targets exacerbates this issue [41]. Additionally, the system’s recognition error for standing behavior was 3.0%, for drinking behavior 0.4%, and for lying down behavior 1.0%. For a 2 h online video, manual processing requires downloading, slowing down, or repeated viewing, whereas this system can collect and record goat behavior information in real time, saving a significant amount of time compared to manual methods. Therefore, the online behavior recognition system for Liaoning Cashmere Goats can achieve relatively accurate behavioral records while greatly reducing the manual labor involved.

### 6.7. Summary of System Construction

This chapter describes the design and construction of an online behavior recognition system for Liaoning Cashmere Goats. The system was designed by dividing it into four main modules: video acquisition, video processing, behavior recognition, and human–machine interaction. A behavior recognition record sheet was created for the goats, and the maximum discrepancy between the behavior records obtained in real time by the system and those obtained by manual review was found to be 6.3%. Consequently, this system can be applied to the context of online behavior recognition for Liaoning Cashmere Goats.

## 7. Conclusions

This paper has developed a lightweight object detection algorithm based on YOLOv8n for the construction of an online detection system for Liaoning Cashmere Goat behavior recognition. We proposed three lightweight schemes for YOLOv8n and integrated data augmentation techniques, CBAM attention mechanisms, and Alpha-CIOU to construct a Liaoning Cashmere Goat behavior recognition algorithm with smaller parameters and higher accuracy, which was then trained. To build an accurate and responsive system, we structured three parts of the experiment:We trained the Liaoning Cashmere Goat behavior recognition dataset using the commonly used object detection algorithms Faster-RCNN and the YOLO series, obtaining the YOLOv8n algorithm with the fastest convergence speed and highest detection accuracy as the ultimate algorithm for behavior recognition.We made lightweight improvements to the feature extraction network of the algorithm using the SlimNeck and GhostNet structures and innovatively proposed a method that combines the two, resulting in the GSNA-YOLOv8n model, which exceeds other SOTA detection algorithms in recognition accuracy. The experimental results of this part indicate that the vertical and horizontal stitching method designed in this study is effective for the behavior recognition dataset of housed goats; the combination of the CBAM attention mechanism and the Alpha-CIOU loss calculation method can compensate for the impact of lightweight convolution on detection accuracy and improve it.We built an online detection system for Liaoning Cashmere Goat behavior recognition, designed a behavior recognition record sheet, and compared the system’s recognition results with the results obtained by manual repeated viewing. The results show that the system constructed in this study can run stably and can replace manual behavior recording for Liaoning Cashmere Goats.

The advantages of the Liaoning Cashmere Goat behavior recognition and online detection system built in this paper are as follows:The model has smaller parameters, occupies less memory, has higher detection accuracy, and specifically addresses the impact of occlusion and dark environments on the annotation of the dataset and the training accuracy of the model.The system includes three models to choose from, high-precision (GSCA-YOLOv8n), lightweight (GNCA-YOLOv8n), and fast (SNCA-YOLOv8n), with both image recognition and video recognition capabilities.After the camera is connected to the system, it can record the behavior of the goats in real time and form reports, providing data support for subsequent behavioral research and daily management.By analyzing the behavior of the goats recorded by the system, it is possible to identify issues such as flock huddling for warmth and gathering in cool areas due to environmental factors during the breeding process; additionally, through the analysis of goat ethology, it is possible to grasp the timing of group feeding and drinking, optimizing the length of feeding troughs and the number of water outlets.This can thereby improve the welfare of the goats.

Due to the constraints of breeding patterns and equipment levels, the degree of intelligence in animal husbandry has just begun, especially the intelligence level of goats, which is far behind that of pigs and chickens. This study is the first to apply machine vision technology to recognize the behavior of Liaoning Cashmere Goats. However, there is still a broad space for development and potential for improvement in the field of deep learning and animal behavior recognition. Future research can delve into the following aspects. First, our current research focuses more on the overall behavioral data analysis of the goatflock, and further accumulation is needed for the abnormal behavior dataset of individual goats. Second, the current algorithm can be combined with target tracking algorithms to recognize the continuous behavior of goats for disease warning and other tasks.

In summary, this study has laid the foundation for technological progress in the field of Liaoning Cashmere Goat behavior recognition and monitoring. Future work will focus more on the optimization and upgrading of models and systems, as well as their extensive application and promotion in actual breeding production, continuing to promote the development of the breeding industry towards intelligence and refinement.

## Figures and Tables

**Figure 1 animals-14-03197-f001:**
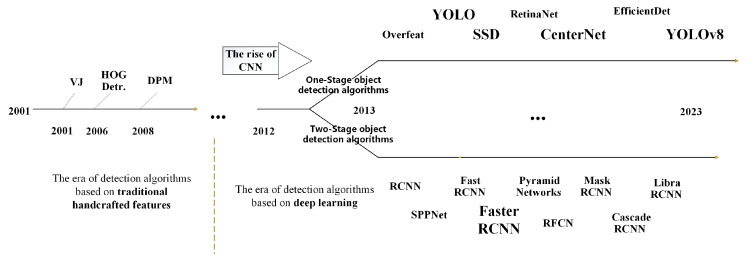
The development history of object detection algorithm.

**Figure 2 animals-14-03197-f002:**
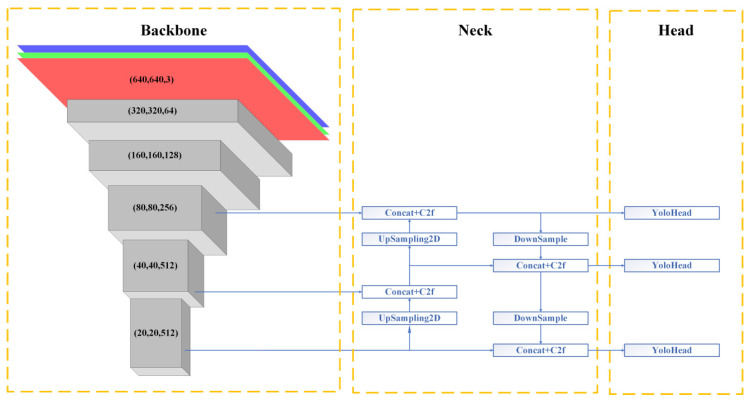
The network architecture diagram of YOLOv8.

**Figure 3 animals-14-03197-f003:**
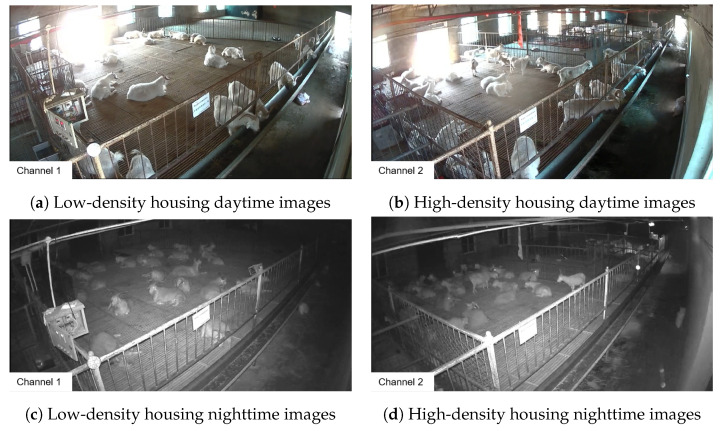
Dataset of goathousing under various rearing densities and lighting coDnditions.

**Figure 4 animals-14-03197-f004:**
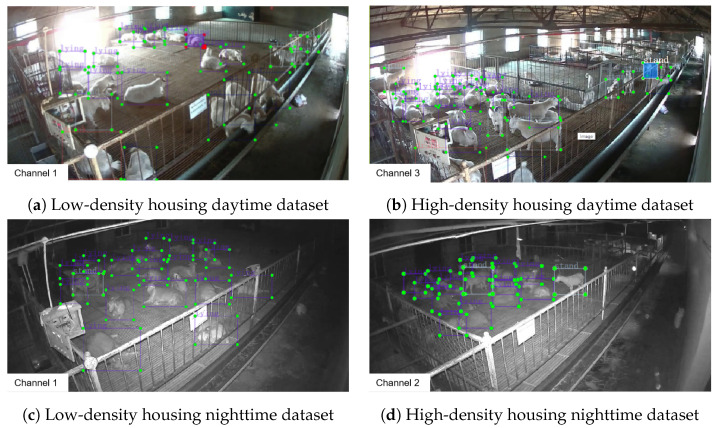
Behavior recognition dataset of Liaoning Cashmere Goats.

**Figure 5 animals-14-03197-f005:**
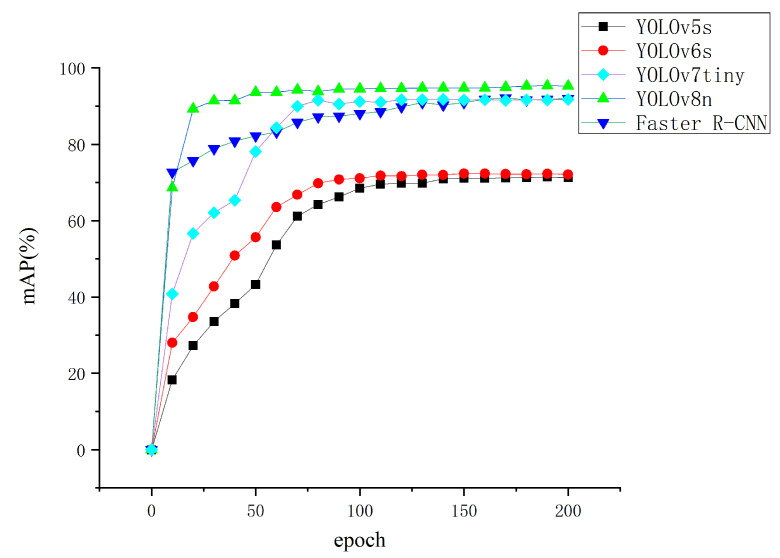
Point-line graph comparing training codes of Faster R-CNN and YOLO series.

**Figure 6 animals-14-03197-f006:**
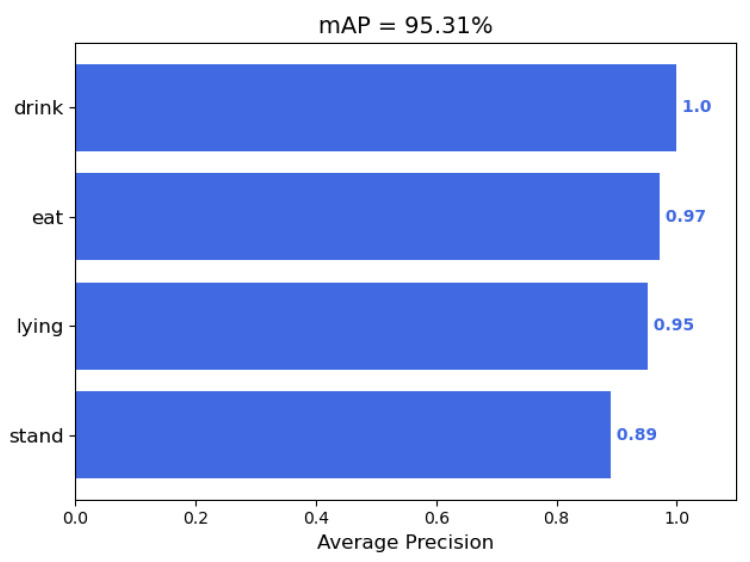
The mAP of YOLOv8n of each behavior trained after 200 epochs.

**Figure 7 animals-14-03197-f007:**
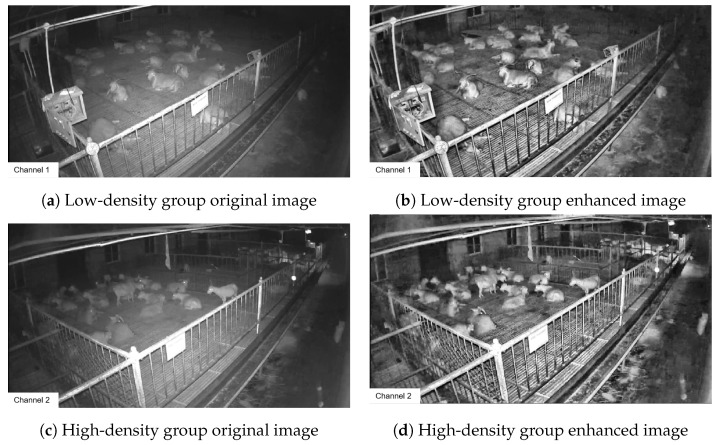
Effect of data augmentation on images of Liaoning Cashmere Goats housed in sheds during nighttime.

**Figure 8 animals-14-03197-f008:**
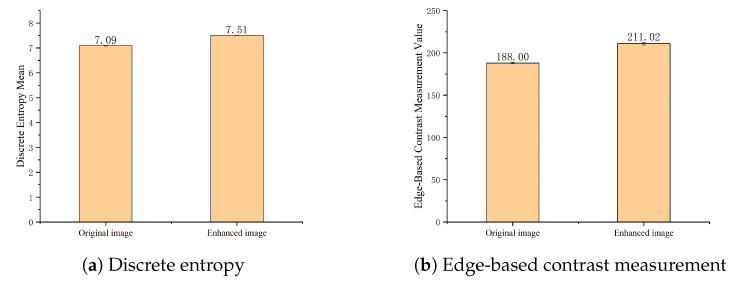
Bar Charts Comparing discrete entropy and edge-based contrast measurement values of nighttime image sets before and after CLAHE enhancement.

**Figure 9 animals-14-03197-f009:**
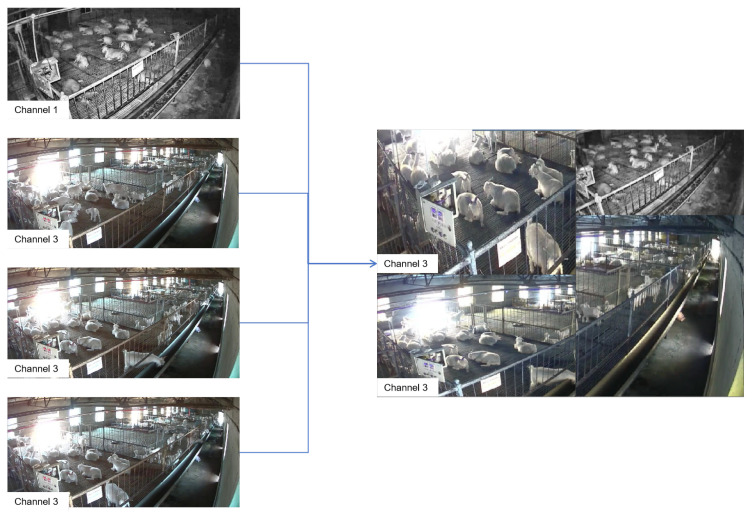
Schematic diagram of the mosaic data augmentation method.

**Figure 10 animals-14-03197-f010:**
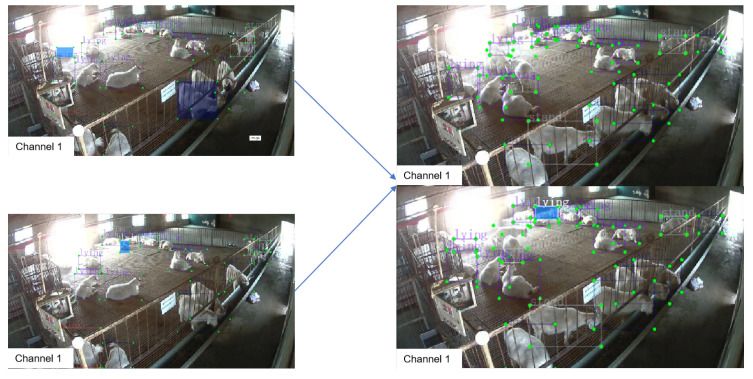
Schematic diagram of data augmentation by vertically combining images.

**Figure 11 animals-14-03197-f011:**
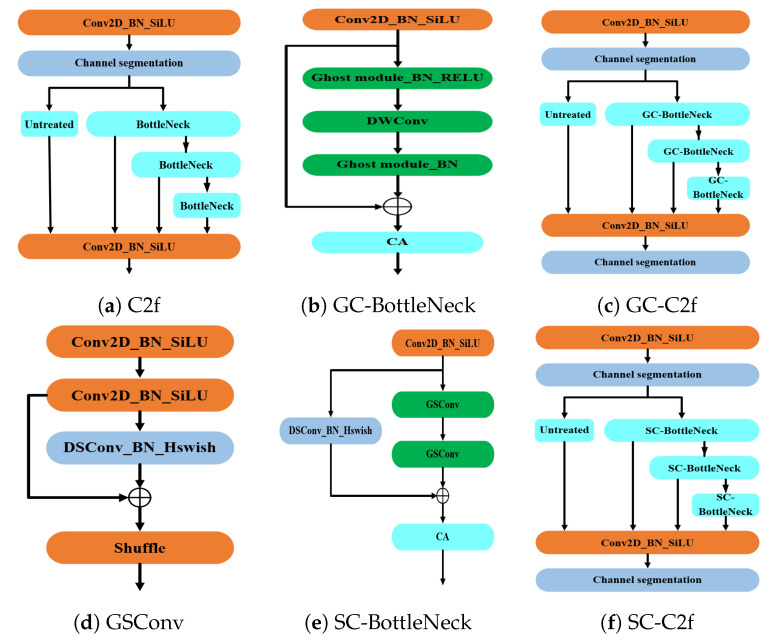
Network structures process of GC-C2f-and SC-C2f.

**Figure 12 animals-14-03197-f012:**
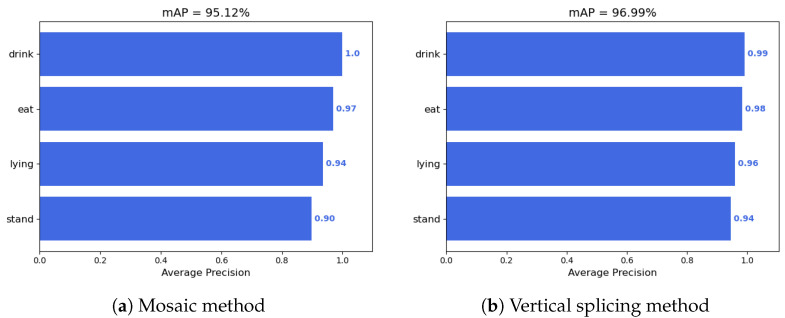
Bar chart of data augmentation results for Mosaic and vertical splicing methods.

**Figure 13 animals-14-03197-f013:**
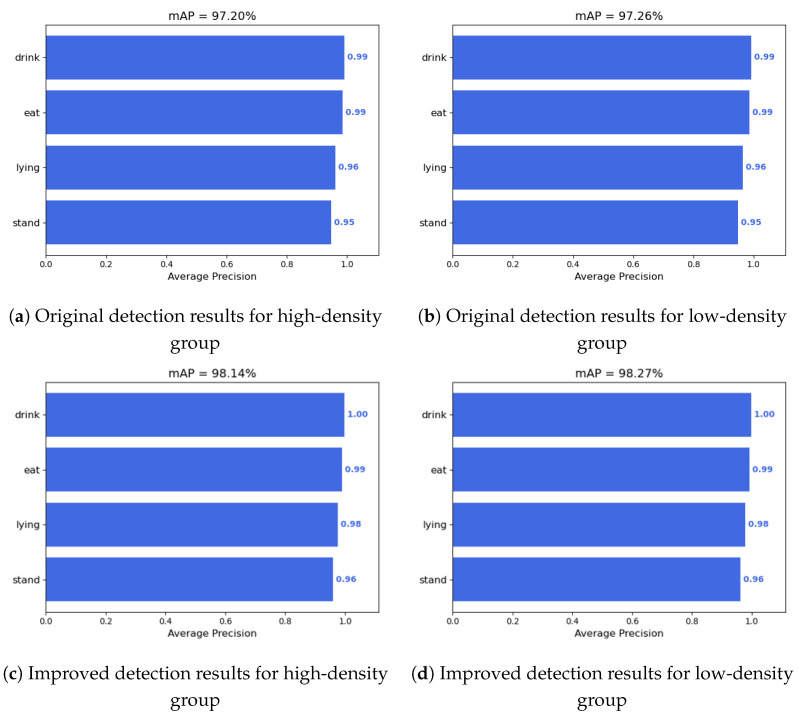
Model comparison plots of mAP in different density groups.

**Figure 14 animals-14-03197-f014:**
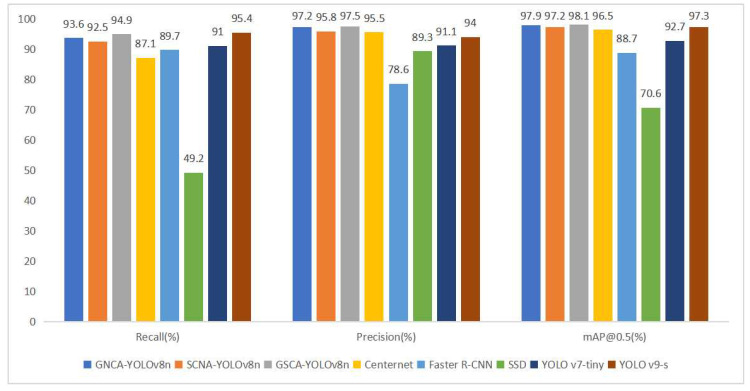
Mosaic schematic diagram of the data enhancement.

**Figure 15 animals-14-03197-f015:**
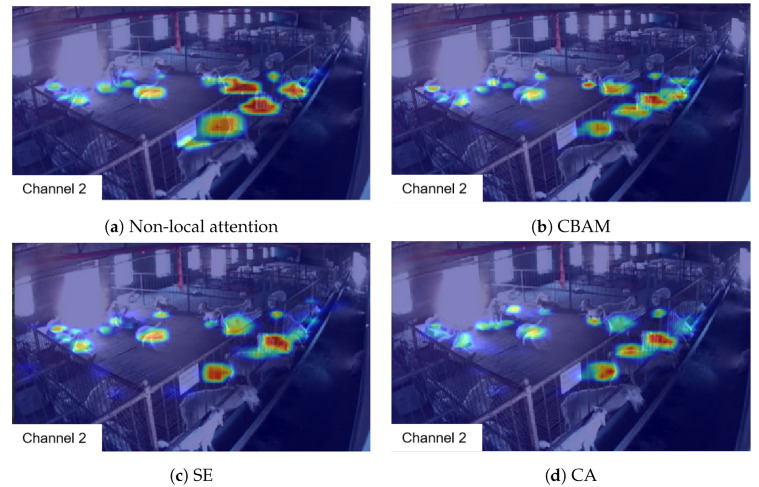
Heatmap of the effects of different attention mechanisms in the FPN. A heatmap is a graphical representation used to illustrate the distribution of a particular variable or set of variables across a space. In the context of this figure, the heatmap represents the intensity of attention focus applied by different attention mechanisms within the Feature Pyramid Network (FPN), visualizing how these mechanisms highlight different regions within an image.

**Figure 16 animals-14-03197-f016:**
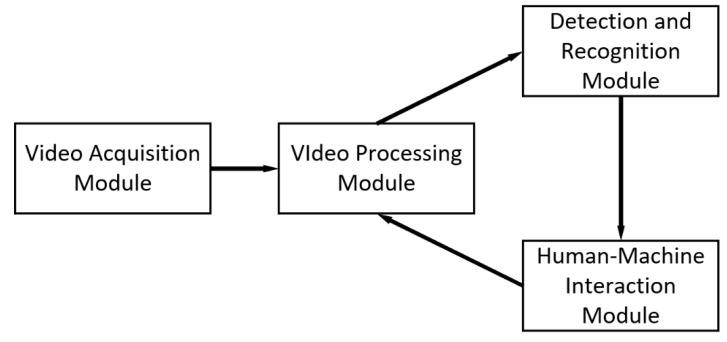
System composition diagram.

**Figure 17 animals-14-03197-f017:**
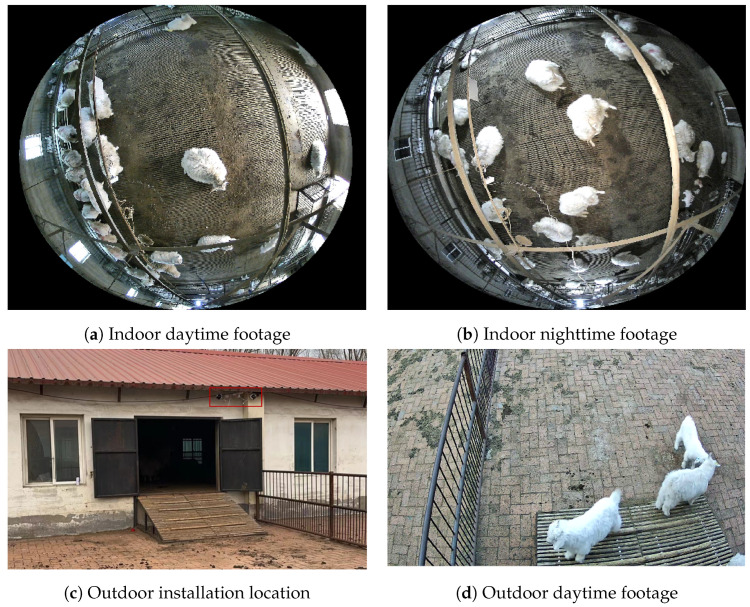
Camera footage from different times and positions inside and outside the shed.

**Figure 18 animals-14-03197-f018:**
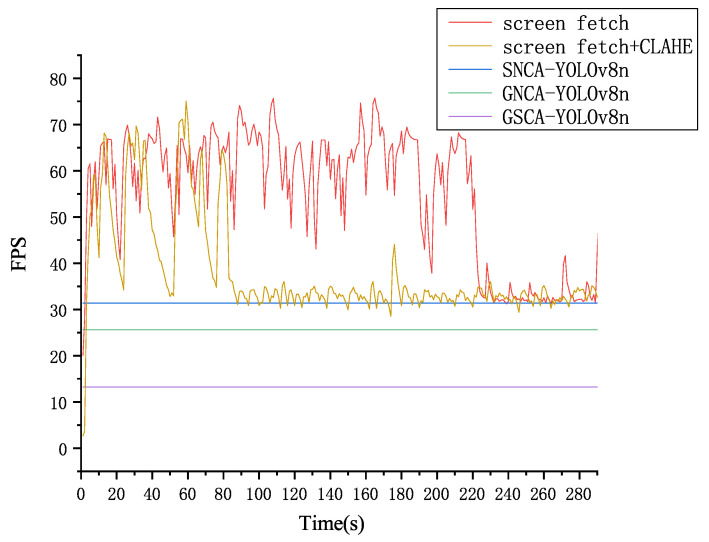
Comparison of video flow frame rate before and after CLAHE.

**Figure 19 animals-14-03197-f019:**
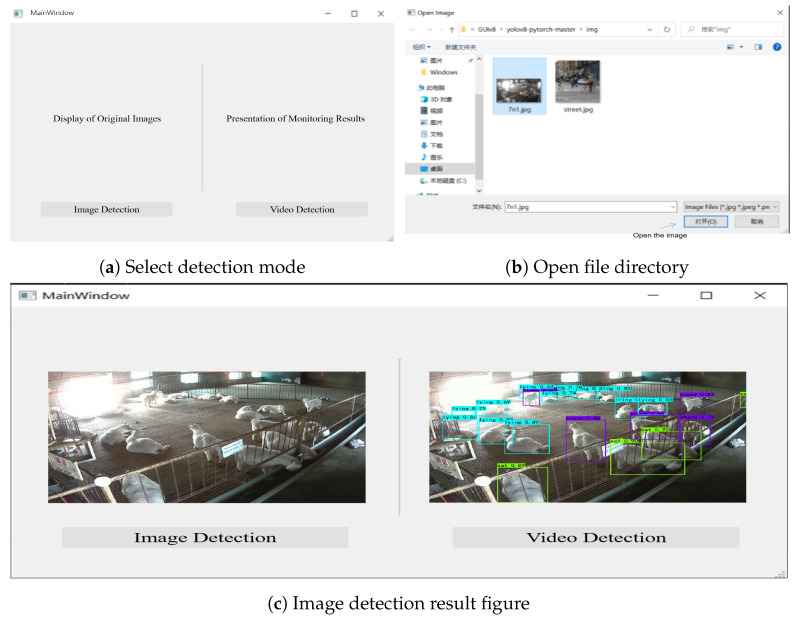
Human–Computer interaction interface.

**Figure 20 animals-14-03197-f020:**
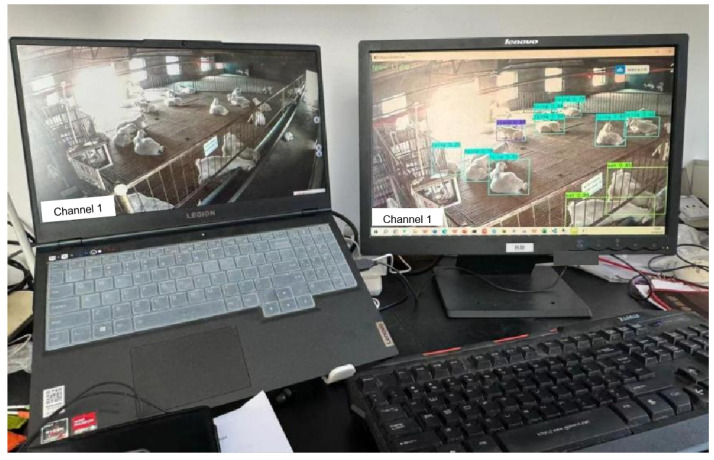
Online identification test of Liaoning Cashmere Goat behavior.

**Figure 21 animals-14-03197-f021:**
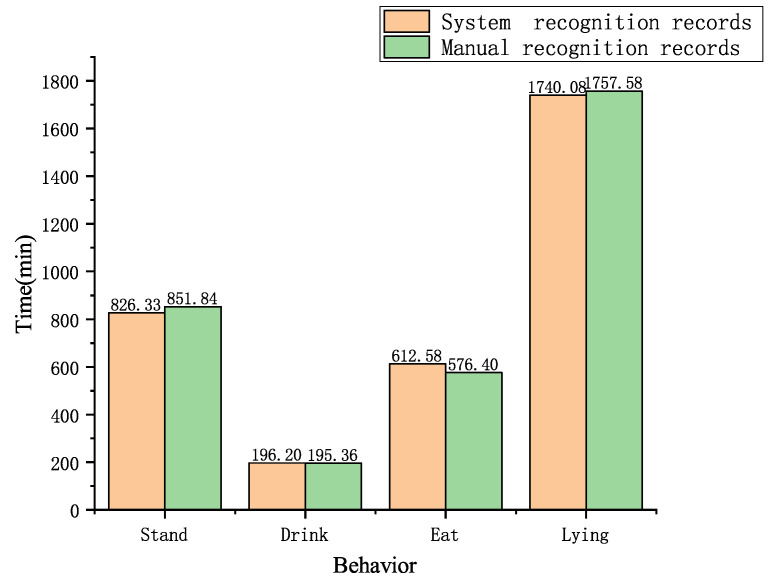
Comparative analysis of system recognition and manual recognition results.

**Table 1 animals-14-03197-t001:** Hardware configuration and experimental environment.

Category	Configuration	Environment	Version
Operating System	Windows 10	Python	3.8
CPU	Intel® Xeon® Gold 6130	PyTorch	1.8.0+cu111
GPU	NVIDIA GeForce RTX 2080 Ti	CUDA	11.7

**Table 2 animals-14-03197-t002:** Improvement results of attention mechanisms and loss functions.

Model	mAP (%)	Memory Usage (MB)	Parameter Count (M)
YOLOv8n	96.99	11.935	3.02
YOLOv8n + SlimNeck	95.34	9.639	2.41
YOLOv8n + SC-C2f + CA	95.84	9.793	2.44
YOLOv8n + SC-C2f + Non-Local Attention	96.17	11.226	2.80
YOLOv8n + SC-C2f + SE	96.10	9.784	2.44
YOLOv8n + SC-C2f + CBAM	96.14	9.838	2.46
YOLOv8n + SC-C2f + CBAM + EIOU	91.98	9.838	2.46
YOLOv8n + SC-C2f + CBAM + SIOU	95.44	9.838	2.46
YOLOv8n + SC-C2f + Non-Local Attention + Alpha-CIOU	97.22	11.180	2.95
YOLOv8n + SC-C2f + SE + Alpha-CIOU	97.02	9.738	2.44
SNCA-YOLOv8n	97.21	9.838	2.46

**Table 3 animals-14-03197-t003:** Results of ablation experiments.

Model	Step	mAP (%)	Memory Usage (MB)	Parameter Count (M)
GNCA-YOLOv8n	step0	96.99	11.94	3.02
	step1	95.72	8.47	2.16
	step2	97.40	8.90	2.21
	step3	97.85	8.90	2.21
SNCA-YOLOv8n	step0	96.99	11.94	3.02
	step1	95.48	9.69	2.42
	step2	96.14	9.84	2.46
	step3	97.21	9.84	2.46
GSCA-YOLOv8n	step0	96.99	11.94	3.02
	step1	95.81	8.49	2.29
	step2	97.61	9.16	2.34
	step3	98.11	9.16	2.34

**Table 4 animals-14-03197-t004:** Comparison of mAP for different behaviors.

Behavior Type	mAP (%)			
	GNCA-YOLOv8n	SNCA-YOLOv8n	GSCA-YOLOv8n	YOLOv8n
Lying	97.35	96.85	97.56	95.98
Stand	95.57	94.25	95.88	94.05
Drink	99.59	99.43	99.86	99.19
Eat	98.92	98.32	99.13	98.35
All Behaviors	97.85	97.21	98.11	96.99

**Table 5 animals-14-03197-t005:** Comparison of memory occupied by different models and frames transmitted per second.

Model	Memory Occupied (MB)	FPS	Parameter Count (M)
GNCA-YOLOv8n	8.90	25.60	2.21
SNCA-YOLOv8n	9.84	31.38	2.36
GSCA-YOLOv8n	9.16	13.24	2.34
YOLOv7-tiny	23.90	22.82	3.11
YOLOv9-s	20.4	29.41	9.6
SSD	94.35	52.17	4.10
Centernet	127.93	34.67	32.72
Faster R-CNN	110.89	15.24	28.36

**Table 6 animals-14-03197-t006:** Specifications of wireless panoramic network cameras and outdoor full-color wireless ball machines (TP-LINK).

Product Feature	Wireless Panoramic Network Camera	Outdoor Full-Color Wireless Ball Machine
Resolution	6 Megapixels	3 Megapixels
Video Transmission Method	WiFi/Ethernet	WiFi/4G/Ethernet
Video Viewing Method	APP/PC Client/URL Streaming	APP/PC Client/URL Streaming
Built-in Storage	128 GB	128 GB
Night Vision Mode	Full-Color Night Vision	Full-Color Night Vision
Weight	0.22 kg	0.372 kg
Dimensions	134 mm × 134 mm × 51 mm	167.5 mm × 116 mm × 156.5 mm

**Table 7 animals-14-03197-t007:** Behavior data records of Liaoning Cashmere Goats.

Date	Time	Standing	Drinking	Lying Down	Feeding
8 October 2023	09:15:03	8	1	10	3
8 October 2023	09:15:04	8	2	9	3
8 October 2023	09:15:05	9	2	9	2
8 October 2023	09:15:06	9	1	9	3
8 October 2023	09:15:07	8	0	11	3
8 October 2023	09:15:08	8	0	12	2
8 October 2023	09:15:09	9	0	12	1
8 October 2023	09:15:10	9	0	12	1
8 October 2023	09:15:11	8	0	12	2
…	…	…	…	…	…
8 October 2023	09:20:04	7	0	13	2

## Data Availability

The datasets used or analyzed during the current study are available from the corresponding author on reasonable request. The datasets generated during the current study are available from the corresponding author upon reasonable request, with the exception of the original video recordings due to intellectual property and patent concerns. These materials are not publicly available but may be accessible for non-commercial research purposes upon negotiation and with proper agreement regarding the terms of use. The code used for data analysis and modeling is available upon request to any qualified researcher for non-commercial purposes, subject to necessary permissions and agreements. Original video recordings are considered part of the materials and are similarly restricted.

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
