# Peer review of "Research on Behavior Recognition and Online Monitoring System for Liaoning Cashmere Goats Based on Deep Learning"

_animals, 2024, doi:10.3390/ani14223197_

Round 1
Reviewer 1 Report
Comments and Suggestions for Authors
See the attachment.

good
Author Response
Comments 1: In Section 3.2.6, Figure 5. Why is it that for each model, evaluation on the test set is not done after one epoch but after every ten epochs?
Response 1: Thank you for your insightful question. The decision to analyze data at multiples of 10 epochs, rather than after each individual epoch, was deliberate and based on several considerations. Firstly, studies such as "Keeping Deep Learning Models in Check: A History-Based Approach" and "Enhancing deep neural network training efficiency and performance through linear prediction" have also opted to evaluate model performance at specific, strategic epochs (e.g., 20, 50, 70, 100, 140). This approach aligns with established research practices. Secondly, the analysis presented in our paper focuses on comparing the early convergence rates and final detection accuracies of different models. The overall trend of the data provides a clear basis for comparison, making the detailed results at every single epoch less significant. Lastly, continuous evaluation after each epoch can significantly slow down the training process, which is detrimental to the expeditious conduct of experiments. By evaluating at every 10 epochs, we strike a balance between monitoring progress and maintaining training efficiency. We believe this approach provides a reasonable trade-off between the granularity of data collection and the practical constraints of training deep learning models.
Comments 2: In Figure 7, more sets of pictures obtained from different channels for image enhancement should be included to more intuitively show the night enhancement effect.
Response 2: Thank you very much for your suggestion to include images that demonstrate the enhancement effects on nighttime images. Following your advice, this paper now includes figures showing the enhancement effects on nighttime images of different densities in the barn. The relevant modifications can still be found in Figure 7 of the paper.
Comments 3: In Figure 10, the advantages of vertical splicing should be explained in detail.
Response 3: Thank you very much for your suggestion to provide a detailed analysis of the results from the vertical stitching method. The relevant modifications have been reflected in lines 335 and 342 of the manuscript, where the changes highlight the shortcomings of the Mosaic data augmentation and the advantages of the vertical stitching enhancement method.
Comments 4: The detection effect diagrams of the algorithm before and after improvement should be provided in different scenarios and briefly explained.
Response 4: Thank you very much for your suggestion to include detection results from different density scenarios before and after the improvements. I have added the detection results for different density pens before the code improvements. This will make my research more comprehensive.
Comments 5: The latest SOTA algorithm comparison lacks the latest YOLOv9, YOLOv11, etc. Please add them.
Response 5: Your suggestion to compare the improved code with the latest YOLO code is crucial. However, due to the higher hardware and software environment requirements for the latest YOLOv10 and YOLOv11, as well as the timing of our research and the availability of data, further exploration is needed on how to better apply them in the field of livestock and poultry. This paper has included the training results of YOLOv9 in the SOTA algorithm comparison chart to ensure that the article keeps up with the cutting-edge research. We plan to investigate the application of YOLOv10 and YOLOv11 in future work.

Reviewer 2 Report
Comments and Suggestions for Authors
Comments and Suggestions
1. This paper addresses a relevant and timely issue in intensive livestock breeding, particularly for the high-value Liaoning Cashmere goats.
2. While you mention that the system improves welfare standards, consider elaborating more on the direct impact of the system on the goats' well-being. For example, how does timely behavior recognition prevent stress, injury, or disease?
3. Since dataset quality and diversity are crucial in deep learning, it would strengthen your study to provide more details on the dataset expansion process. What types of behaviors were added? How was the data collected, and did you address any specific challenges in labeling or augmenting the data?
4. Explain the approach behind making YOLOv8n more lightweight. This might involve reducing the number of parameters, using model pruning, or optimizing the architecture for hardware constraints. Discuss how these changes helped improve real-time detection without compromising accuracy.
5. It would be beneficial if you would make a comparison of your methods with other systems for different animals such as cows. In this aspect, I would like to refer you to the following for references.
6. Su Myat Noe, Thi Thi Zin, Pyke Tin, I. Kobayashi, "Comparing State-of-the-Art Deep Learning Algorithms for the Automated Detection and Tracking of Black Cattle", Sensors, 2023, 23(1), 532
7. Thi Thi Zin, I. Kobayashi, Pyke Tin, H. Hama, “A General Video Surveillance Framework for Animal Behavior Analysis”, Proceedings - 2016 3rd International Conference on Computing Measurement Control and Sensor Network, CMCSN 2016, 2017, pp. 130–133, 8008657.
Author Response
Comments 1: While you mention that the system improves welfare standards, consider elaborating more on the direct impact of the system on the goats' well-being. For example, how does timely behavior recognition prevent stress, injury, or disease?
Response 1: Thank you for pointing this out. I agree with this comment. Therefore, I have added a discussion on the impact of the system on animal welfare at lines 654-659 of the manuscript and highlighted it in red.
Comments 2: Since dataset quality and diversity are crucial in deep learning, it would strengthen your study to provide more details on the dataset expansion process. What types of behaviors were added? How was the data collected, and did you address any specific challenges in labeling or augmenting the data?
Response 2: I completely agree with your statement regarding the importance of dataset creation and expansion in object detection. The dataset used in this study was collected from the Liaoning Cashmere Goat Breeding Center, where a substantial amount of actual production monitoring videos were obtained. After acquiring the videos and extracting key frames, the behaviors of the goats in the images were categorized into four classes: "stand," "lying," "eat," and "drink," followed by annotation. During the annotation process, it was observed that nighttime images posed challenges due to low lighting, making annotation difficult; thus, data augmentation was applied to these nighttime images. Subsequently, a comparison was made between the Mosaic data augmentation method and the vertical stitching data augmentation method in terms of principles and results, concluding that the vertical stitching method was superior. Therefore, the final dataset used for training was determined.
Comments 3: Explain the approach behind making YOLOv8n more lightweight. This might involve reducing the number of parameters, using model pruning, or optimizing the architecture for hardware constraints. Discuss how these changes helped improve real-time detection without compromising accuracy.
Response 3: Thank you very much for your suggestions regarding the lightweight design of the algorithm. Following your advice, I have made revisions to lines 409-422 of the manuscript, with the relevant content highlighted in red. I have ensured a detailed analysis of the impact of lightweight design on model parameter count and accuracy mechanisms.
Comments 4: It would be beneficial if you would make a comparison of your methods with other systems for different animals such as cows. In this aspect, I would like to refer you to the following for references.
Response 4: Thank you very much for your suggestion to compare this system with other systems for different animals, such as cows. I have revised lines 601-602 of the manuscript based on the literature you provided, and the changes have been highlighted in red.
